# Diversity and prevalence of zoonotic infections at the animal-human interface of primate trafficking in Peru

A. Patricia Mendoza[1,2,3¤]*, Ana Muñoz-Maceda[4], Bruno M. Ghersi[5], Micaela De La Puente[1], Carlos Zariquiey[1], Nancy Cavero[1], Yovana Murillo[1], Miguel Sebastian[1], Yohani Ibañez[1], Patricia G. Parker[2], Alberto Perez[6], Marcela Uhart[7], Janine Robinson[4], Sarah H. Olson[8], Marieke H. Rosenbaum[5]

**1** Wildlife Conservation Society - Peru Program, Lima, Peru, **2** Department of Biology, University of Missouri - Saint Louis, St Louis, Missouri, United States of America, **3** Asociación Neotropical Primate Conservation – Perú, Moyobamba, San Martín, Perú, **4** School of Anthropology and Conservation, Durrell Institute of Conservation and Ecology, University of Kent, Canterbury, Kent, United Kingdom, **5** Department of Infectious Disease and Global Health, Cummings School of Veterinary Medicine at Tufts University, North Grafton, Massachusetts, United States of America, **6** Servicio Nacional de Sanidad y Calidad Agroalimentaria, Buenos Aires, Argentina, **7** One Health Institute, University of California - Davis, Davis, California, United States of America, **8** Wildlife Conservation Society - Health Program, Bronx, New York, United States of America

¤ Current address: Department of Anthropology, Washington University in St. Louis, St Louis, Missouri, United States of America

* anapatricia.mendoza@gmail.com

**Editor:** Érika Martins Braga, Universidade Federal de Minas Gerais, BRAZIL

**Data Availability Statement:** The data that support the findings of this study are openly available in figshare at http://doi.org/10.6084/m9.figshare. 21363183.

## Abstract

Wildlife trafficking creates favorable scenarios for intra- and inter-specific interactions that can lead to parasite spread and disease emergence. Among the fauna affected by this activity, primates are relevant due to their potential to acquire and share zoonoses - infections caused by parasites that can spread between humans and other animals. Though it is known that most primate parasites can affect multiple hosts and that many are zoonotic, comparative studies across different contexts for animal-human interactions are scarce. We conducted a multi-parasite screening targeting the detection of zoonotic infections in wild-caught monkeys in nine Peruvian cities across three contexts: captivity (zoos and rescue centers, n = 187); pet (households, n = 69); and trade (trafficked or recently confiscated, n = 132). We detected 32 parasite taxa including mycobacteria, simian foamyvirus, bacteria, helminths, and protozoa. Monkeys in the trade context had the highest prevalence of hemoparasites (including *Plasmodium malariae/brasilianum*, *Trypanosoma cruzi*, and microfilaria) and enteric helminths and protozoa were less common in pet monkeys. However, parasite communities showed overall low variation between the three contexts. Parasite richness (PR) was best explained by host genus and the city where the animal was sampled. Squirrel (genus *Saimiri*) and wooly (genus *Lagothrix*) monkeys had the highest PR, which was ~2.2 times the PR found in tufted capuchins (genus *Sapajus*) and tamarins (genus *Saguinus/Leontocebus*) in a multivariable model adjusted for context, sex, and age. Our findings illustrate that the threats of wildlife trafficking to One Health encompass exposure to multiple zoonotic parasites well-known to cause disease in humans, monkeys, and other species. We demonstrate these threats continue beyond the markets where wildlife is

**Funding:** This study was made possible thanks to the generous support of the American people through the United States Agency for International Development (USAID) Emerging Pandemic Threats Program – PREDICT (cooperative agreement number GHN-A-OO-09-00010-00). MU contribution to this manuscript is supported by award U01AI151814 from the National Institute of Allergy and Infectious Diseases of the National Institutes of Health. MR contribution to this research was partially funded by National Institutes of Health Office of the Director, Fogarty International Center, Office of AIDS Research, National Cancer Center, National Eye Institute, National Heart, Blood, and Lung Institute, National Institute of Dental & Craniofacial Research, National Institute On Drug Abuse, National Institute of Mental Health, National Institute of Allergy and Infectious Diseases Health, and NIH Office of Women's Health and Research through the International Clinical Research Fellows Program at Vanderbilt University (R24 TW007988) and the American Relief and Recovery Act, as well as by the National Center for Advancing Translational Sciences, National Institutes of Health, Award Number UL1TR001064. The contents are the responsibility of the authors and do not necessarily reflect the views of USAID, the National Institutes of Health, or the United States Government. The funders had no role in study design, data collection and analysis, decision to publish, or preparation of the manuscript.

**Competing interests:** The authors have declared that no competing interests exist.

initially sold; monkeys trafficked for the pet market remain a reservoir for and contribute to the translocation of zoonotic parasites to households and other captive facilities where contact with humans is frequent. Our results have practical applications for the healthcare of rescued monkeys and call for urgent action against wildlife trafficking and ownership of monkeys as pets.

## Introduction

In order for a parasite to spread from one host species to another, both species have to co-occur within the same geographical and ecological boundaries [1–5]. In nature, species co-occurrence is determined by geographic overlap of their niche and home range, which in turn dictates the environmental and ecological conditions under which they live [4,6]. Wildlife trade introduces novel anthropogenic parameters that influence parasite dispersal and host-range. For example, trade routes and husbandry practices govern the geographic range, intra- and inter-species interactions, and level of exposure of wildlife to humans at live-animal markets, zoos, rescue centers, and households [7–9]. Wildlife trade is considered an important driver of parasite spread and disease emergence because it facilitates parasite sharing between species and individuals that do not naturally interact with each other, such as humans and most wildlife species [10–13]. To note a few examples, wildlife trade and traffic, wildlife markets, and wildlife pets have been associated with the global spread of chytrid fungus [14], the 2003 outbreak of Monkeypox in the United States [15], the H5N1 Avian Influenza [16,17] and SARS [18] epidemics, and presumably the recent SARS-CoV-2 pandemic [19]. Of highest concern among infections that can spread through wildlife trade and traffic are the zoonoses - those infections caused by parasites that can be transmitted between humans and other animals.

Primates are among the most trafficked wildlife species, and as described above, this trade facilitates parasite sharing between humans and other primates species that would otherwise be excluded from urban environments [20]. As a result of their interaction with us, captive non-human primates (NHP) acquire human-associated parasites that make their microbiome and parasite communities more similar to those of humans than to those of their wild counterparts [21,22]. Non-human primates are also susceptible to infectious disease caused by human-associated parasites such as tuberculosis, herpesvirus, and influenza virus, which can cause important morbidity and mortality in free-ranging apes and monkeys [23–27]. Despite the intense trade of non-human primates, their high potential to carry and spread zoonotic parasites, and the associated risk of spillover to free-ranging populations, the breadth of infections supported by primate trafficking is not well described.

The Neotropics, particularly the Amazon basin, harbor the highest richness of primate species around the world [28]. This region is also a hotspot for wildlife trafficking [28–30] and tropical diseases [31,32]. Currently, there are no facilities in South America authorized to breed monkeys for the pet market. Consequently, pet monkeys from this region are obtained through illegal hunting in their natural tropical forest habitats, which also represent areas of high endemism for tropical diseases of public health relevance [9,33]. In Peru alone, the illegal pet trade affects over 30 monkey species [34]. Monkeys offered for sale or obtained illegally as pets are frequently confiscated by local authorities, temporarily held in custody facilities, and ultimately euthanized or transferred to government-regulated zoos and rescue centers [34,35]. This illicit trade facilitates animal-human interactions and disease exposure, spanning from

the forests where monkeys are hunted, through markets and trade networks, to households keeping monkeys as pets illegally, and the captive facilities where confiscated monkeys are housed [9]. Through trafficking, monkeys are also transported to regions outside their natural habitats or areas where specific tropical diseases may be absent, but where ecological and environmental factors are favorable for disease vectors and infectious agents, thereby raising additional concerns regarding the potential introduction and spillover of diseases [36–39].

More than 320 parasite taxa have been reported in monkeys across the Americas, encompassing various types such as enteric helminths, enteric protozoa, hemoparasites, viruses, bacteria, and ectoparasites [40–44]. Among these, at least 74 taxa are known to infect humans [42,43]. Focusing on zoonotic or potentially zoonotic parasites, studies in Peruvian free-ranging and captive monkeys have detected infections with hemoparasites belonging to the genera *Trypanosoma*, *Plasmodium*, *Dipetalonema*, and *Mansonella* [45–51], enteric helminths such as *Ancyclostoma* sp., *Ascaris* sp., *Strongyloides spp.*, *Trypanoxiuris sp.*, *Trichuris trichiura*, and *Schistosoma mansoni.* [52–55], and enteric protozoa of the genera *Blastocytis*, *Balantidium*, *Cryptosporidium*, *Coccidia*, *Eimeria* and *Entamoeba* [54–57]. In the northern Peruvian Amazon, the zoonotic bacteria *Campylobacter spp.* were reported in 21% wild monkeys and 32% pet monkeys [58], while primate rescue centers in the same region report primates infected with antimicrobial-resistant enterobacteria such as *Escherichia coli*, *Salmonella arizonae*, *Shigella sp.*, *and Serratia spp.* [41]. Within the country, human-associated infections such as mycobacteria [59] and human herpesvirus type 1 [60] have been documented exclusively in captive primates, whereas a high seroprevalence of arbovirus antibodies has been observed solely in wild monkeys [61,62]. These zoonotic or potentially zoonotic parasites are often opportunistic, capable of infecting multiple host species, and are commonly transmitted through arthropod vectors or easily acquired from environments contaminated with secretions and excreta.

Among captive monkeys, the risk of infection may be higher in areas where vector ecology is similar to their natural habitat, in wildlife markets and facilities where hygiene and sanitary conditions are poor, and within the crowded and confined spaces where monkeys are kept or offered for sale. These conditions are more commonly encountered during the initial stages of trafficking [63–65]. On the other hand, the negative impact of zoonotic diseases on both captive and free-ranging monkey populations [24,66–68] prompts government-regulated facilities like zoos and rescue centers to implement health assessments, disease screening, and preventive medicine for monkeys recovered from wildlife trafficking or illegal pet ownership [27,35]. These measures should reduce parasite burden in captive collections and pre-release facilities, thereby improving the welfare of rescued monkeys, and mitigating disease risks for those exposed to them. Nevertheless, the lack of comprehensive knowledge regarding the spectrum of parasites carried by trafficked monkeys makes the prevention of disease and spillover difficult.

Assessing the variation in parasite communities across different captive settings can illustrate the breadth of infections harbored by monkeys introduced to human-inhabited areas. Such information has practical applications in the identification of disease risk, prevention of zoonotic threats, and guidance of conservation efforts to safeguard One Health - the intertwined health of humans, non-human animals, and ecosystems [69]. Between 2010-2013, the Emerging Pandemic Threats PREDICT program in Peru contributed to this effort by demonstrating that zoonotic parasites circulate across the country via wildlife trafficking [9,51,59,70,71]. Here, we evaluate the diversity and prevalence of these infections across the animal-human interface of monkey trafficking in Peru. We hypothesize that the risk of parasite detection in captive monkeys is higher in wildlife markets than in other contexts in which the animals are found. We also assess contributing factors to parasite richness to identify

geographic hotspots and species most likely to host parasites capable of infecting and causing disease in humans and other animals.

## Methods

### Study design and sampling strategy

Between September 2010 through May 2012, we conducted a cross-sectional study with opportunistic collection of blood, saliva, and faecal samples from captive monkeys (Parvorder *Platyrrhini*) in nine Peruvian cities (Fig 1). We sampled wild-caught monkeys at zoos, rescue centers, households, wildlife markets, and temporary custody facilities and classified them into three distinct 'contexts' where animal-human interaction occurs: 1) captivity (zoos and rescue centers); 2) pet (households); and 3) trade (wildlife markets or seized during transport to commercial establishments). Monkeys confiscated by local authorities or voluntarily surrendered were assigned to the context they came from (trade or pet) and sampled within the first week of their arrival at the custody facility. To ensure that only animals originated in the primate trafficking were represented in the study, monkeys born in captivity were excluded. Aggressive, debilitated, highly stressed, and nursing monkeys were also excluded. Consent from monkey caretakers was obtained prior to sample collection.

### Samples collection and processing

For the collection of samples, monkeys under 2.5 kg were physically restrained for no longer than 10 minutes, whereas larger monkeys were chemically restrained using a Xylazine - Ketamine - Midazolam protocol aiming for 30 minutes sedation. A 1-3 ml blood sample was collected by venipuncture of the femoral vein, not exceeding 0.6% of the animal's body weight in grams, placed in EDTA collection tubes, gently homogenized and aliquoted as follows: one to three drops were placed in dry-blood spot cards (Whatman FTA Classic cards, and Whatman 903 Protein Saver cards), two drops were used to produce a thin and thick blood smear, and the remaining whole blood was placed in cryogenic tubes. Saliva was collected by gently swabbing the oral mucosa with a sterile polyester swab and placed in lysis buffer and universal transport media as described by Rosenbaum et al. [59]. Whole blood and oral swabs were maintained in refrigeration during field collection and stored at -80˚C until laboratory analysis was conducted. Faecal samples were obtained through rectal swabbing and scooping of fresh droppings. Rectal swabs were placed in tubes containing Cary-blair media and maintained in refrigeration until their arrival to the laboratory for immediate processing. Approximately 0.5 grams of fresh faeces were placed in sodium acetate formalin (SAF), and 1-2 grams were placed in 2% phosphate-buffered saline solution (PBS). Both faecal aliquots were maintained at room temperature until their arrival to the laboratory for immediate processing. Laboratory tests aimed for direct detection of infectious agents through light microscopy, bacteriological culture, and PCR. Sampling and parasite detection procedures are summarized in Table 1.

### Statistical analysis

All the infectious agents we tested for are referred to as 'parasites' because the presence of clinical disease was not investigated [77]. Parasite prevalence was estimated as the number of infected individuals over the total number of individuals evaluated for each parasite taxa and parasite type (i.e., mycobacteria, viruses, hemoparasites, enteric bacteria, enteric helminths, enteric protozoa, and trichomonads). We tested the homogeneity of the frequency of each parasite type among the three contexts using Pearson's Chi-squared test ($\alpha = 0.05$) and compared prevalence using 95% Wald confidence intervals with continuity correction.

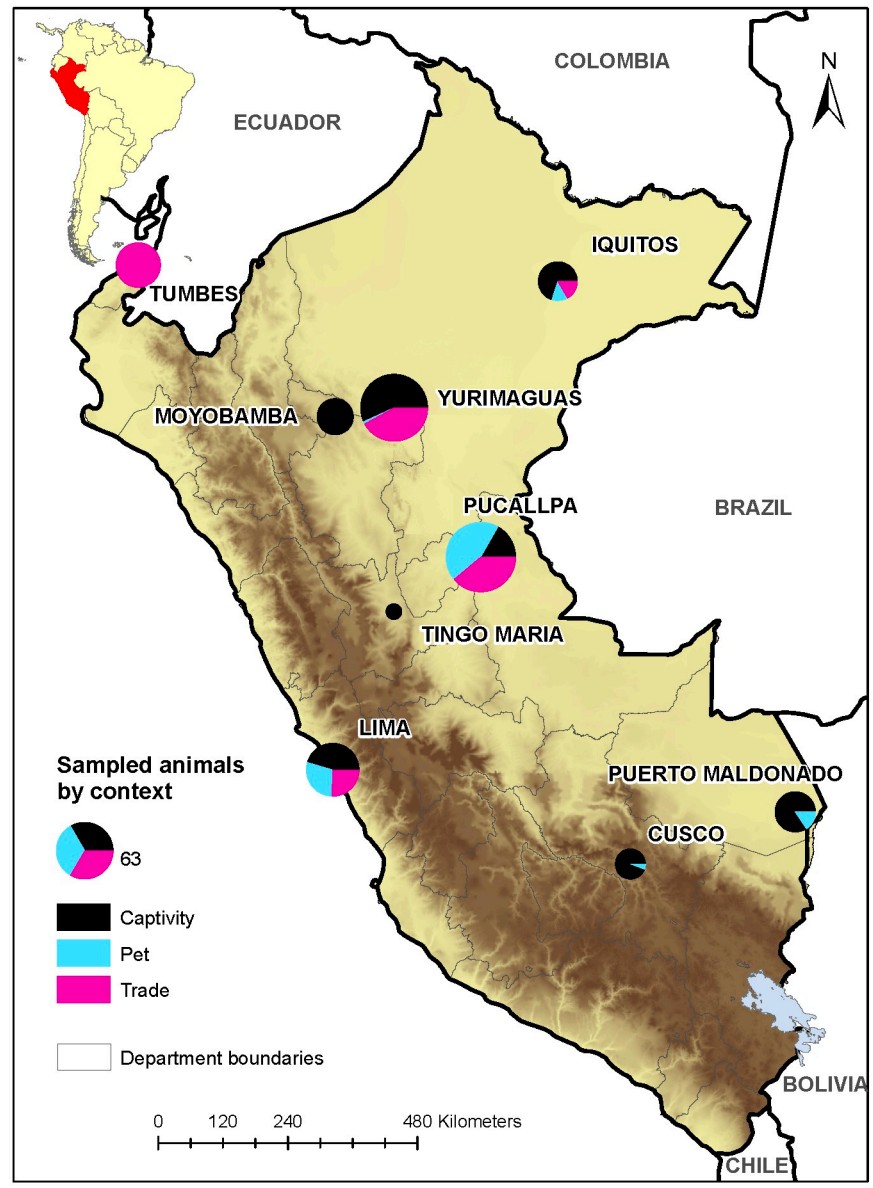

**Fig 1. Map of Peru showing the distribution of the study population by context and city.** Pie charts are proportional to the number of monkeys sampled in each city, whether at government-regulated captive facilities (Captivity, black), at households (Pet, light blue), or at markets (Trade, pink). Insert shows the location of Peru in South America. See Table 2 for further details.

Parasite richness (PR) was estimated as the number of parasite taxa detected in each monkey by context (captivity, pet, trade), host genera, sex, age category (infant, juvenile, adult), and location (city where the monkey was sampled). Due to changing primate taxonomy, the

**Table 1. Sampling, storage, and laboratory technique used for the detection of zoonotic parasites in captive, pet, and traded monkeys in Peru.**

| Test<br>Parasite taxa detected | Type of sample | Storage medium | Laboratory technique | Reference* |
|---|---|---|---|---|
| *1. Mycobacterium tuberculosis* complex (MTBC) molecular detection | | | | |
| MTBC | Oral swab | Lysis Buffer | PCR IS6110 | [59] |
| 2. Simian Foamyvirus (SFV) serological and molecular detection | | | | |
| SFV | Blood | DBS card (PS903) | EIA-WB, PCR | [71] |
| 3.Human T-lymphotropic virus (HTLV) serological and molecular detection | | | | |
| *None* | Blood | DBS card (PS903) | EIA-WB, PCR | [71] |
| 4.Influenza A/B molecular detection | | | | |
| *None* | Oral swab | UTM | RT-PCR | [72] |
| 5.Microfilaria detection | | | | |
| *Dipetalonema sp.* | Thin & thick blood smear | 2% formalin/none | Light microscopy, Knott's test | [73] |
| *Mansonella sp.* | | | | |
| 6.Trypanosomatid detection | | | | |
| *Trypanosoma sp. (excluding T.cruzi)* | Blood smear | None | Light Microscopy | [73] |
| | Blood | EDTA, DBS card (FTA) | PCR D71 & D75 | [51] |
| *Trypanosoma cruzi* | Blood smear | None | Light Microscopy | [73] |
| | Blood | EDTA, DBS card (FTA) | PCR D71 & D75 | [51] |
| 7.Malaria detection | | | | |
| *Plasmodium malariae/P.brasilianum* | Blood smear | None | Light Microscopy | [73] |
| | Blood | EDTA, DBS card (FTA) | PCR 18s rRNA | [74] |
| 8.Enteric bacteria detection | | | | |
| *Aeromonas caviae* | Faeces | Cary Blair | Bacterial culture, isolation, and serotyping | [75] |
| *Aeromonas hydrophila* | | | | |
| *Aeromonas sobria* | | | | |
| *Aeromonas sp. (excluding A. caviae, A. hydrophila, and A. sobria)* | | | | |
| *Campylobacter coli* | | | | |
| *Campylobacter jejuni* | | | | |
| *Campylobacter sp. (excluding C. jejuni and C. coli)* | | | | |
| *Salmonella O Group D* | | | | |
| *Shigella boydii* | | | | |
| *Shigella flexnerii* | | | | |
| *Shigella sonnei* | | | | |
| *Plesiomonas shigelloides* | | | | |
| 9.Faecal parasites detection | | | | |
| *Hookworm* | Faeces | SAF/None | Light microscopy | [76] |
| *Molineus sp.* | | | | |
| *Prosthenorchis sp.* | | | | |
| *Strongyloides sp.* | | | | |
| *Trichuris sp.* | | | | |
| *Ascaris sp.* | | | | |
| *Balantidium sp.* | | | | |
| *Blastocystis sp.* | | | | |
| *Entamoeba sp.* | Faeces | SAF/None | Light microscopy, PCR | |
| *Cryptosporidium sp.* | | | | |
| *Giardia sp.* | | | | |

*(Continued)*

**Table 1.** (Continued)

| Test<br>Parasite taxa detected | Type of sample | Storage medium | Laboratory technique | Reference* |
|---|---|---|---|---|
| 10.Trichomonads detection | | | | |
| *Dientamoeba sp.* | Faeces | PBS | Light microscopy | [76] |
| *Trichomonas sp.* | | | | |

DBS: Dry blood spot; PS903: Whatman 903 Protein saver card, FTA: Whatman FTA card, UTM: Universal transport media; EDTA: Blood collection tubes embedded with ethylenediaminetetraacetic acid; SAF: sodium-actetate formalin; PBS: Phosphate-buffered saline solution; PCR: Polymerase chain reaction, EIA-WB: Enzymatic immunoassay – Western blot, RT-PCR: Real-time polymerase chain reaction.

* Further details about sample processing and laboratory techniques can be found in the reference.

**Table 2. Sampling effort and characteristics of the study population.**

| | N | Animal-human context | | | MPD<br>Median (IQR) | PR |
|---|---|---|---|---|---|---|
| | | Captivity (N = 187) | Pet (N = 69) | Trade (N = 132) | | |
| **Sampling events** | | | | | | |
| Number of events | 106 | 25 | 54 | 34 | | -- |
| Mean group size (min, max) | 4 (1,42) | 7 (1,41) | 1(1,3) | 4(1,37) | | -- |
| **Taxonomic family** | | | | | | |
| *Aotidae* | 12 | 4 | 4 | 4 | 17 (12.75-18.75) | 6 |
| *Atelidae* | 161 | 123 | 28 | 10 | 18 (6-28) | 24 |
| *Callitrichidae* | 44 | 9 | 6 | 29 | 13 (12-14) | 8 |
| *Cebidae* | 162 | 48 | 28 | 86 | 17 (13-22) | 25 |
| *Pitheciidae* | 9 | 3 | 3 | 3 | 17 (13-24) | 4 |
| **Common name**<br>***Taxonomic genus (species, sample size)*** | | | | | | |
| Howler monkeys<br>*Alouatta (A. seniculus*, n = 17*)* | 17 | 9 | 6 | 2 | 17 (17-28) | 8 |
| Owl monkeys<br>*Aotus (A. nigriceps*, n = 1*; Aotus sp.*, n = 11*)* | 12 | 4 | 4 | 4 | 17 (12.75-18.75) | 6 |
| Spider monkeys<br>*Ateles (A. belzebuth*, n = 8*; A. chamek*, n = 27*)* | 35 | 28 | 6 | 1 | 17 (13-28) | 13 |
| Uakaris<br>*Cacajao (C. calvus*, n = 3*)* | 3 | 1 | 2 | 0 | 24 (23.5-24.5) | 2 |
| Pygmy marmosets<br>*Callithrix (C. pygmaea*, n = 5*)* | 5 | 0 | 1 | 4 | 13 (13-15) | 3 |
| Gracile capuchins<br>*Cebus (Cebus sp.*, n = 25*)* | 25 | 13 | 6 | 6 | 17 (13-17) | 12 |
| Woolly monkeys<br>*Lagothrix (L. flavicauda*, n = 1*, L. lagotricha*, n = 108*)* | 109 | 86 | 16 | 7 | 18 (6-26) | 21 |
| Saki monkeys<br>*Pithecia (Pithecia sp.*, n = 3*)* | 3 | 0 | 1 | 2 | 13 (8-18.5) | 1 |
| Titi monkeys<br>*Plecturocebus (P. cupreus*, n = 1*; P. oenanthe*, n = 2*)* | 3 | 2 | 0 | 1 | 17 (10-17) | 2 |
| Tamarins<br>*Leontocebus/Saguinus (Leontocebus sp.*, n = 36*;*<br>*S. mystax*, n = 3,*)* | 39 | 9 | 5 | 25 | 13 (12-14) | 6 |
| Squirrel monkeys<br>*Saimiri (S. boliviensis*, n = 9*; S. cassiquiarensis*, n = 45*)* | 94 | 14 | 7 | 73 | 18 (13-18) | 21 |

*(Continued)*

**Table 2.** (Continued)

| | N | Animal-human context | | | MPD Median (IQR) | PR |
|---|---|---|---|---|---|---|
| | | Captivity (N = 187) | Pet (N = 69) | Trade (N = 132) | | |
| **Sampling events** | | | | | | |
| Tufted capuchins *Sapajus* (*S. macrocephalus*, n = 43) | 43 | 21 | 15 | 7 | 17 (13-26) | 12 |
| **Sex** | | | | | | |
| Female | 174 | 101 | 25 | 48 | 17 (13-24.75) | 30 |
| Male | 181 | 82 | 42 | 57 | 17 (13-24) | 28 |
| Unknown | 33 | 4 | 2 | 27 | 12 (12-13) | -- |
| **Age category** | | | | | | |
| Infant | 46 | 13 | 11 | 22 | 20 (13-24) | 22 |
| Juvenile | 149 | 50 | 35 | 64 | 16 (13-24) | 29 |
| Adult | 177 | 121 | 21 | 35 | 17 (13-26) | 25 |
| Unknown | 16 | 3 | 2 | 11 | 12.5 (1.25-14.25) | -- |
| **City** | | | | | | |
| Cusco | 18 | 17 | 1 | 0 | 17 (17-17) | 8 |
| Iquitos | 30 | 21 | 4 | 5 | 13 (13-13.75) | 4 |
| Lima | 55 | 25 | 16 | 14 | 18 (13-28) | 18 |
| Moyobamba | 26 | 26 | 0 | 0 | 20 (13.75-28) | 10 |
| Puerto Maldonado | 32 | 27 | 5 | 0 | 17 (17-24) | 13 |
| Pucallpa | 95 | 16 | 42 | 37 | 14 (7.5-24) | 20 |
| Tingo María | 5 | 5 | 0 | 0 | 23 (23-23) | 6 |
| Tumbes | 39 | 0 | 0 | 39 | 12 (12-13) | 13 |
| Yurimaguas | 88 | 50 | 1 | 37 | 18 (13-28.25) | 21 |

Sampling events, number of monkeys evaluated (N) at each context for animal-human interaction, maximum possible detection (MPD), and parasite richness (PR) by family, genus, sex, age, and city.

IQR: Interquartile range.

genera *Leontocebus* and *Saguinus* were clustered and analyzed as the *Leontocebus/Saguinus* group. Host genera with fewer than 25 individuals were grouped together and analyzed as 'Other'. Monkeys sampled in the same day at the same location were considered part of the same 'sampling event'. The number of parasite taxa detected by each test was used to estimate the 'maximum possible detection' (MPD). MPD = the number of parasite taxa that could have been detected by all the tests carried out on samples from the same monkey. We used Pearson's product-moment correlation to assess the effect of group size (number of individuals per sampling event) and MPD in PR.

Differences in parasite community composition were evaluated using a presence/absence matrix of parasite genera detected by context and host genera. This matrix was used to estimate the multi-assemblage Sorensen coefficient as a measure of dissimilarity. Accounting for differences in parasite richness, we partitioned these estimates to determine the amount of dissimilarity due to parasite turnover and to nestedness [78,79]. To visualize parasite community similarity, we used the same presence/absence matrix and carried out a principal components analysis with singular value decomposition. We retained the first two principal components (PC) to represent the variance in parasite community composition in two dimensions. Factor loadings were calculated as the correlation between the selected PC and the presence/absence of each parasite genera by context and host genus.

Generalized linear models (GLMs) with a negative binomial error distribution were built to evaluate the relationship between PR and the attributes of the study population, including host genus, age category, sex, context, and location as predictors, and maximum possible detection as an offset. Alternative generalized linear mixed effect models (GLMMs) were built using the same predictors as fixed terms and sampling event as a random term. Model selection utilized stepwise term deletion, using the Akaike's information criterion corrected for small sample sizes (AICc) to identify the best models. Adjusted incidence rate ratios (IRR) between categories were calculated as the exponentiated coefficients of the model. All statistical analyses were performed using R 4.0.3 (R Development Core Team, 2021), RStudio (RStudio Team, 2023) the *vegan* [80], the *DescTools* [81], the *betapart* [79], the *factoextra* [82], the *glmulti* [83], and the *lmer4* [84] packages.

### Ethics statement

This research was authorized by the Peruvian government through permit N˚0411-2010-AG-DGFFS-DGEFFS and N˚ 0618- 2011-AG-DGFFS-DGEFFS for the project "Infectious diseases in the wild animal trade in Peru". The procedures for animal sampling were evaluated by the Institutional Committee for Animal Use and Care of the School of Veterinary Medicine of the University of California, Davis, and approved under IACUC Protocol WCS-PREDICT #16027.

### Results

We carried out 106 sampling events with a mean group size of four individuals per event (range: 1-42). Samples were obtained from 388 monkeys including 18 species and 12 genera within the five families of the Parvorder Platyrrhini. Large-bodied monkeys (i.e, *Lagothrix*, *Alouatta*, *Ateles*) were more frequent in 'captivity' whereas smaller genera (i.e., *Saimiri*, *Saguinus/Leontocebus*, and *Callithrix*) were more often found in the 'trade' context (Table 2).

Up to seven tests were carried out on samples from the same monkey (median number of tests=3.5, IQR = 2.0-5.0) reaching a maximum possible detection of 31 parasite taxa (median MPD=17, IQR = 13.0-24.0). Given the limitations for bleeding and rectal swabbing in smaller species, fewer tests were carried out on samples collected from marmosets (genus *Callithrix*, median = 2.0, IQR = 2.0-3.0) and tamarins (genus *Saguinus/Leontocebus*, median = 2.0, IQR = 1.0-2.0) compared to other genera (Table 2). A total of 1,313 tests were performed, resulting in the detection of 32 parasite taxa. Nearly half (44%) of these parasites were identified to the species level and are known to infect and cause disease in humans, and those identified to the genus-level corresponded to genera with at least one human-infecting species, with the possible exception of *Prostenorchis sp*. and *Molineus sp*. PCR testing for Influenza A/B (n = 39) and Human T-lymphotropic virus (n = 19) were the only assays with no positive detections (Table 3).

We found a significant difference in the prevalence of hemoparasites [$X^2$=69.7, p<0.001], enteric helminths [$X^2$=22.1, p<0.001], and enteric protozoa [$X^2$=28.8, p<0.001] across contexts (S1 Table). We observed a higher prevalence of hemoparasites and enteric helminths in the 'trade' context whereas enteric helminths and enteric protozoa were less prevalent in 'pet' monkeys (Figs 2 and S2–S6).

Variation in parasite community composition was low when comparing the parasite genera found across contexts (Sorensen dissimilarity coefficient=0.300, 74% turnover). However, parasite communities were more similar between 'trade' and 'pet' (Sorensen dissimilarity coefficient=0.167, 71% turnover) than between 'trade' and 'captivity' (Sorensen dissimilarity coefficient=0.235, 57% turnover) (S2 Table).

**Table 3. Detection (+/-) of parasites in monkeys across different contexts for animal-human interaction generated by wildlife trafficking in Peru.**

| Parasite type | Parasite taxa | Captivity | Pet | Trade | Reported in humans | Reported in free-ranging primates |
|---|---|---|---|---|---|---|
| Mycobacteria | *Mycobacterium tuberculosis* complex | + | + | + | yes | Catarrhines only [85] |
| Virus | Simian foamyvirus | + | - | + | yes | [86] |
| | Influenza A/B | - | - | - | yes | Catarrhines only [23,87] |
| | Human T-lymphotropic virus | - | - | - | yes | --- |
| Hemoparasite | *Dipetalonema sp.* | - | + | + | yes | [49] |
| | *Mansonella sp.* | + | + | + | yes | [49] |
| | *Trypanosoma cruzi* | + | - | + | yes | [51] |
| | *Trypanosoma sp. (exc. T. cruzi)* | + | + | + | yes | [49] |
| | *Plasmodium malariae/brasilianum* | - | - | + | yes | [49] |
| Enteric bacteria | *Aeromonas caviae* | + | - | + | yes | --- |
| | *Aeromonas hydrophila* | - | + | + | yes | --- |
| | *Aeromonas sobria* | - | + | + | yes | --- |
| | *Aeromonas sp. (exc. A.caviae, A. hydrophila, A.sobria)* | + | - | + | yes | [88] |
| | *Campylobacter coli* | + | + | + | yes | [58] |
| | *Campylobacter jejuni* | - | - | + | yes | [58] |
| | *Campylobacter sp. (exc. C. coli, C. jejuni)* | + | - | - | yes | [58] |
| | *Salmonella O Group D* | - | + | - | yes | Catarrhines [89,90] |
| | *Shigella boydii* | - | - | + | yes | Apes [89], platyrrhines[†] [91] |
| | *Shigella flexnerii* | - | - | + | yes | Apes [89], platyrrhines[†] [91] |
| | *Shigella sonnei* | + | - | - | yes | Apes [89], platyrrhines[†] [91] |
| | *Plesiomonas shigelloides* | - | + | - | yes | --- |
| Enteric helminth | *Hookworm* | + | + | + | yes | [92] |
| | *Molineus sp.* | - | - | + | no | [53,92] |
| | *Prosthenorchis sp.* | - | + | + | no | [52,53] |
| | *Strongyloides sp.* | + | + | + | yes | [52,57] |
| | *Trichuris sp.* | + | - | - | yes | [52,57] |
| | *Ascaris sp.* | + | + | + | yes | [52] |
| Enteric protozoa | *Balantidium sp.* | + | - | + | yes | [93] |
| | *Blastocystis sp.* | + | + | + | yes | [56] |
| | *Entamoeba sp.* | + | + | + | yes | [54] |
| | *Cryptosporidium sp.* | + | + | + | yes | [94] |
| | *Giardia sp.* | + | + | + | yes | [95] |
| Trichomonad | *Dientomoeba sp.* | + | NT | + | yes | Apes [96] |
| | *Trichomonas sp.* | + | NT | - | yes | [97] |

NT: Not tested. †*Shigella sp.*

The PC analysis suggested a similar pattern (Fig 3). The first two PCs represented 32.1% and 16% of the variation among contexts and host genera and showed that the parasite community composition in *Ateles*, *Lagothrix*, and *Saimiri* were more similar between 'trade' and 'pet' than when compared with the parasite communities of monkeys of the same genus at the 'captivity' context. PC1 was negatively correlated with most parasite genera but did not have a significant correlation with any bacteria. PC2 was loaded by hemoparasites in the negative end and was positively correlated with trichomonads and *Shigella* (S3 Table).

A maximum of nine parasite taxa were detected in the same individual (mean PR = 1.237, s. d. = 1.654). PR showed a moderate positive correlation with MPD (r = .60, p<0.001) and there was a low correlation between the median PR and event size (r = .23, p = 0.017).

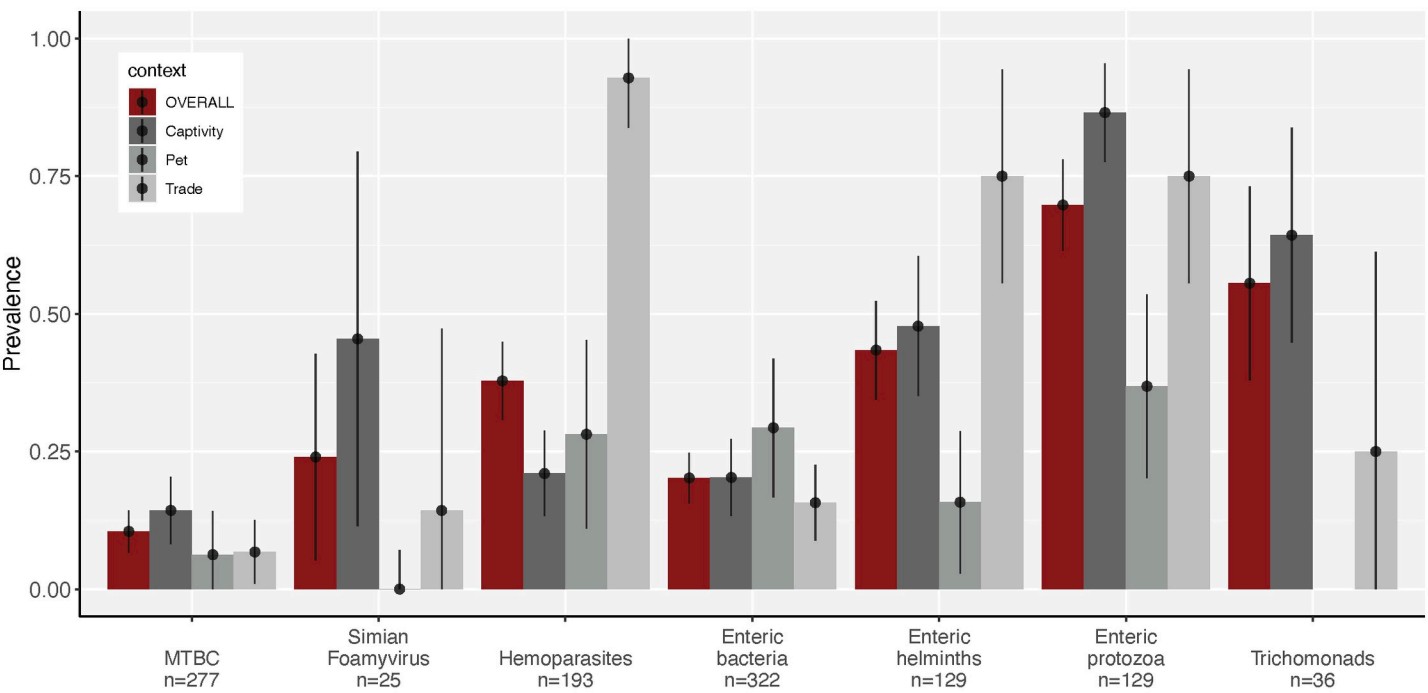

**Fig 2. Prevalence of zoonotic parasites in captive monkeys in Peru.** Bars and dots correspond to the prevalence of each parasite type in the sampled population (red) and among animal-human contexts (grey). Vertical lines indicate 95% confidence intervals. Horizontal lines preceded by an asterisk indicate significant difference between categories (p<0.05). MTBC: *Mycobacterium tuberculosis* complex; n: number of individuals tested for each parasite type.

The most plausible model explaining PR in captive monkeys in Peru included host genus and city, and these were the only two terms that contributed to more than 80% of the models. Age category and sex contributed similarly to the second-best model, when working with alternative data sets including either age category or sex (S4 Table). After adjusting for genus and city, while offsetting MPD, context, age category, and sex did not have an effect on PR (Fig 4). 'Sampling event' was nested within city and its inclusion as a random term did not improve the model (S4 Table).

Among host genera, higher PR was observed in squirrel (genus *Saimiri*) and wooly (genus *Lagothrix*) monkeys (Table 2). Respectively, squirrel and wooly monkeys have an incidence rate ratio 135% (IRR: 2.27, 95%C.I.: 1.38-3.86) and 119% (IRR: 2.19, 95%C.I.: 1.35-3.67) higher than tufted capuchins (genus *Sapajus*), and 129% (IRR: 2.29, 95%C.I.: 1.32-4.24) and 120% (IRR: 2.20, 95%C.I.: 1.26-4.10) higher than tamarins (genus *Saguinus/Leontocebus*) when all other categories were held constant (Fig 4). The PR was significantly higher in Yurimaguas than in other cities, except Moyobamba and Tingo María (Table 2). Moyobamba had a higher PR than Pucallpa (IRR: 2.13, 95%C.I.: 1.25-3.59) and Tumbes (IRR: 1.92, 95%C.I.: 1.04-3.58); and Iquitos had a significantly lower PR than any other city (IRR: 0.19-0.46). Other pairwise comparisons were not significant.

## Discussion

SARS-CoV-2 and other recent epidemics presumably linked to wildlife use and consumption have directed global attention towards the discovery of novel pathogens associated with wildlife trade [98,99]. Our results illustrate that well-known, widespread zoonotic infections can be a daily threat at the animal-human interface created through wildlife trafficking that should be

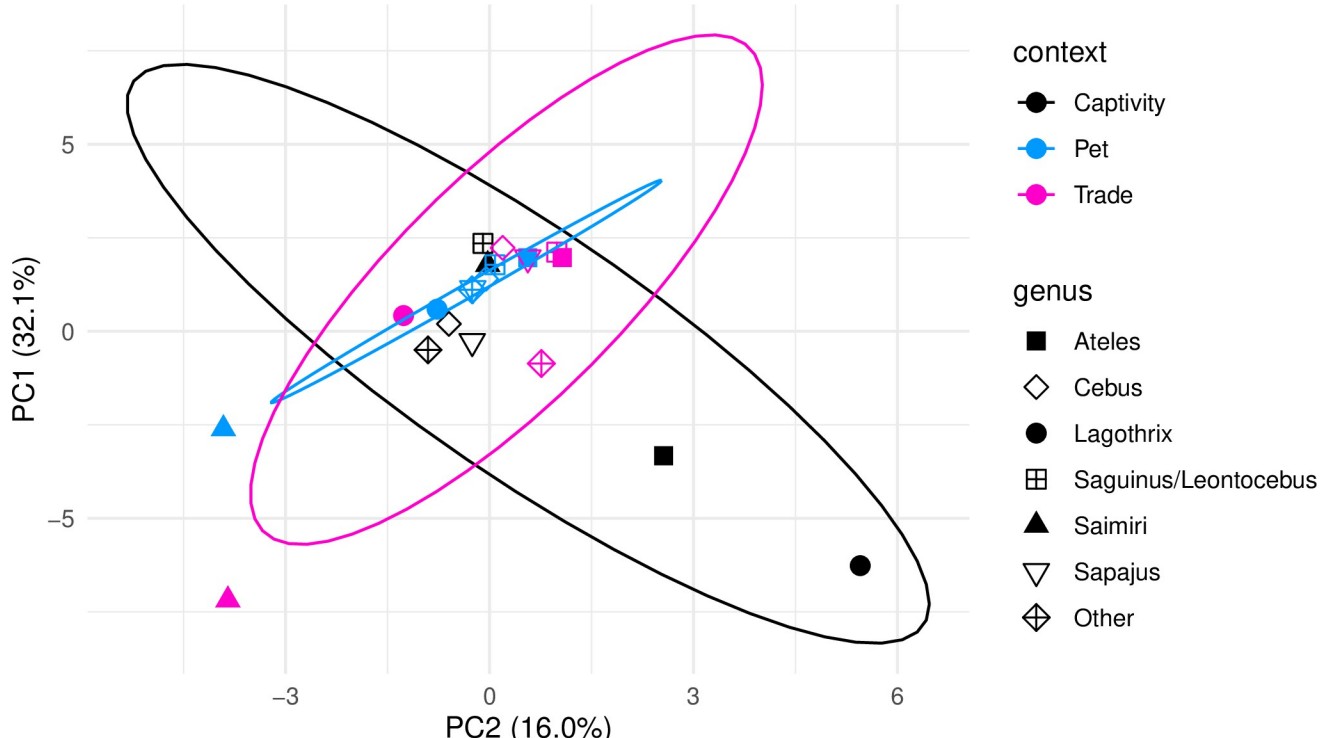

**Fig 3. Parasite community similarities among animal-human contexts.** Principal component (PC) analysis showing the variance in parasite presence among host genera and context in two dimensions. The symbols represent the parasite community of each monkey genus at each animal-human context, and the distance between them illustrates their dissimilarity. Ellipses correspond to the 95% confidence interval for each context.

addressed in parallel to the study of novel pathogens. We detected parasites transmitted by vectors, direct contact, or through exposure to contaminated soil and surfaces, showing there is a constant risk of exposure to zoonotic pathogens for human and animal populations at the places where trafficked monkeys are found.

## Zoonotic infections in trafficked monkeys

We found that exposure to trafficked monkeys from Peru pose risks for the transmission of *Trypanosoma cruzi*, *Plasmodium malariae/brasilianum*, *Mycobacterium tuberculosis* complex (MTBC), and a broad spectrum of enteric parasites. These widespread infections contribute significantly to human morbidity, mortality, and disability in low and middle income countries (LMIC), disproportionally affecting impoverished communities and vulnerable populations without proper access to water and sanitation [100]. The World Health Organization (WHO) lists Chagas disease, caused by *Trypanosoma cruzi*, and soil-transmitted helminthiasis as neglected tropical diseases affecting approximately eight and two million people, respectively [101]. *Plasmodium malariae/brasilianum* and MTBC cause human malaria and tuberculosis respectively, two diseases that took the lives of more than two million people in 2020 alone and currently account for the largest burden of infectious diseases in tropical countries [100]. In addition, *Salmonella*, *Shigella*, and *Campylobacter* are among the most common causes of food-borne human bacterial enteritis, reactive arthritis, and traveler's diarrhea syndrome [102,103]. *Campylobacter* infection is also associated with impaired growth of children in Peru and other LMIC countries [104]. At markets, live monkeys are often observed in the proximity of poultry and raw meat, while pet monkeys are often in close direct contact with

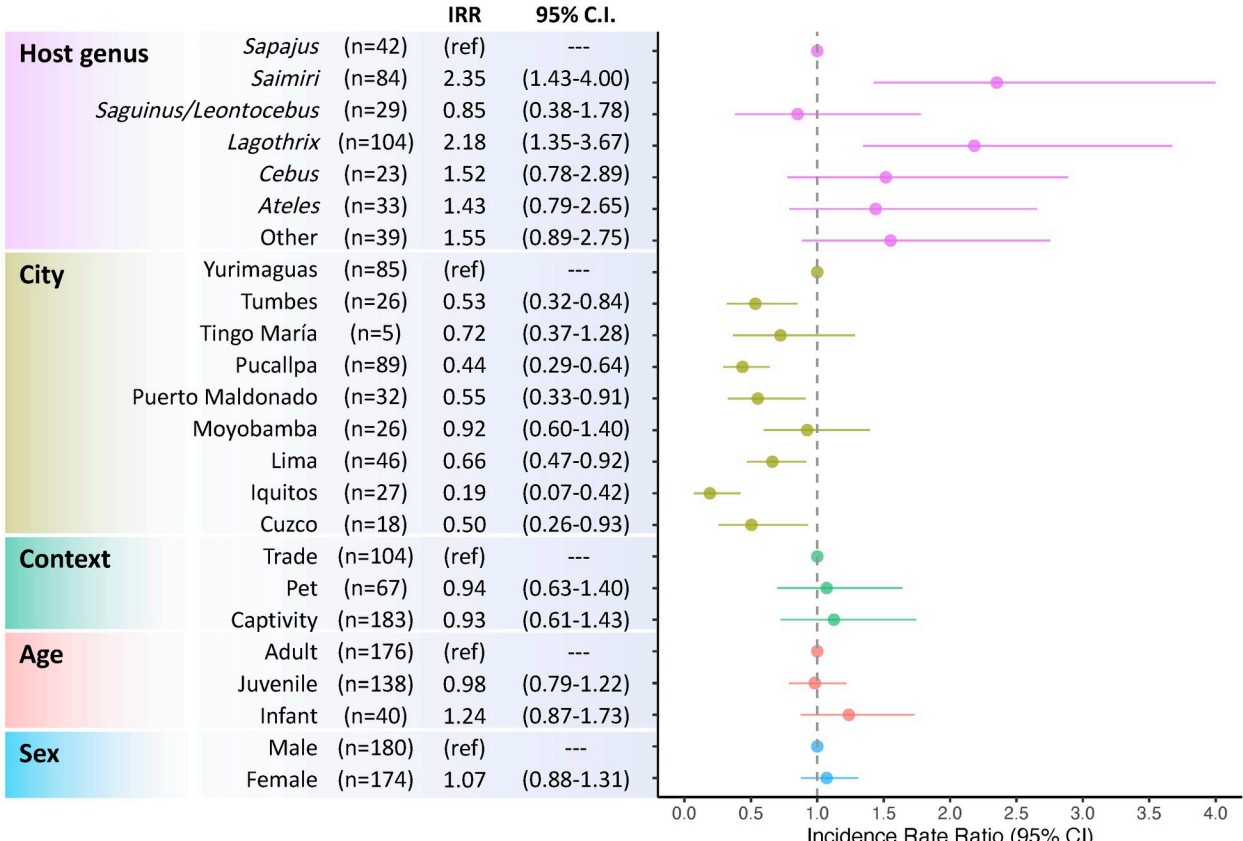

**Fig 4. Risk factors for Parasite Richness (PR) in captive, pet, and traded monkeys in Peru.** Incidence Rate Ratio (IRR) for PR calculated using a negative binomial regression model adjusted by the maximum possible detection of the tests carried on each monkey. IRR: adjusted incidence rate ratio; 95% C.I.: 95% confidence interval; ref: reference group. Dots and lines correspond, respectively, to the adjusted IRR and 95% C.I. of the full model.

children and vulnerable adults within the family environment [8,9]. Although infected monkeys may have acquired zoonotic parasites from humans or other infected animals while in captivity, their presence in markets and households represents an additional source of contamination and exposure to zoonotic pathogens at these scenarios. Documenting how often these infections are acquired from trafficked animals presents an important avenue for future research. It would be essential to explore, for example, the contribution of captive monkeys as reservoirs of *Plasmodium malariae/brasilianum* to the burden of human malaria giving the little information available about this parasite in comparison to malaria agents that are not considered as zoonotic.

There is also a gap in our knowledge of how the presence of zoonotic parasites in a host contributes to disease transmission and emergence in an ecosystem. The impact of the introduction of zoonotic parasites to free-ranging monkey populations has ranged from self-limiting outbreaks and die-offs [23,25,66] to the establishment of sylvatic cycles and further amplification [105,106]. Releasing trafficked and rescued monkeys may expose free-ranging populations to zoonotic parasites, with unknown consequences particularly for endangered

species. Of great concern, MTBC is a group of human-associated pathogens that has not been reported in free-ranging monkeys in the Americas but severely affects platyrrhine monkeys in captivity [68,85,107]. MTBC was also absent in African monkeys until the spillover of bovine tuberculosis in Kruger National Park reached several endangered species, with rapid progression of the disease and up to 50% mortality in chacma baboons (*Papio ursinus*) [108,109]. MTBC introduction to wild settings in Peru could add to the disease burden of already endangered species, but most importantly, its spread to wildlife reservoirs through primate reintroductions could establish the disease in systems where control and eradication would become unfeasible [110,111]. We stress the need for strict enforcement of MTBC testing of captive primates and their caretakers, and the withholding of primate releases in any situation where a negative infection status cannot be confirmed. Similar precautions must be taken with any human- or domestic animal- associated pathogen with potential to harm the health of free-ranging populations.

### Pathogen communities across contexts for animal-human interaction

We found that the context in which captive monkeys are found has little influence on the parasite communities they harbor. We anticipated that initial stages of trafficking would be more favorable for infections, and thus monkeys at markets or recently recovered from wildlife trafficking would bear a higher prevalence and diversity of infections than monkeys at government-regulated captive facilities where disease prevention and treatments are often applied. We found, however, that pathogen communities and PR were similar among the three contexts studied and only hemoparasites and enteric helminths were found at higher prevalence in the 'trade' context.

The vector-borne hemoparasites detected in our sample (genus *Trypanosoma*, *Dipetalonema*, *Mansonella*, and *Plasmodium*,) are endemic to the Amazon forests that serve as sources for wildlife trafficking. Simultaneous circulation of these hemoparasites was reported by Erkenswick et al. [48,49] in free-ranging tamarins in Peru with annual prevalence as high as 100% for *Trypanosoma minasense*, 76% for *Dipetalonema spp*., 67% for *Mansonella mariae*, and 14% for *Plasmodium malariae/brasilianum*. Though the ecology of these infections in monkeys is not fully studied, trypanosomatids and filarial nematodes are well tolerated at high loads and high prevalence by free-ranging Neotropical monkeys [49,112–114], whereas malaria parasites are observed at low prevalence and low parasitemia [115,116]. Monkeys in the 'trade' have been captive for a shorter time than those in other contexts, and thus are more likely to reflect the intensity of infections at their source populations [51], which explains the higher prevalence and diversity of hemoparasites found at this context (Figs 2 and S3). Concurrently, if morbidity of captive monkeys is increased due to hemoparasitic infections and/or transmission within captive facilities is not sustained, hemoparasite prevalence may decrease over time. The lower diversity and prevalence of hemoparasites in 'pet' and 'captivity' suggests that transmission within households, zoos, and rescue centers is infrequent and may not compensate the mortality of infected individuals.

Most of the enteric helminths (Hookworms, *Prostenorchis sp*., *Strongyloides sp*., *Trichuris sp*., *Ascaris sp*.) we found are soil-transmitted and their prevalence is associated with contaminated environments with human and animal feces, and mud or dirt floors [117,118]. Anecdotally, these conditions were met in all the markets we visited, and the highly contaminated environment likely impacts the higher prevalence of enteric helminths observed in the 'trade' context (Figs 2 and S5). Open latrines, yards and cages contaminated with excreta, and dirt floors were also common in households with 'pet' monkeys, but enteric helminths and protozoa were less prevalent at this context (Figs 2 and S5). This difference could be partially

explained by lower parasite loads in household environments where only one to three monkeys are found, in contrast with the hundreds to thousands of animals sold daily at markets [8,34,64]. Although illegally-acquired pets rarely receive veterinary healthcare [119], it is also likely that some pet monkeys are medicated with over-the-counter antihelmintics or antibiotics lowering our detection of parasite shedding. Another plausible explanation is given by the potentially better nutritional status of 'pet' monkeys in comparison with those in the 'trade'. Improved nutrition can increase the resistance to helminth infections and enhance the efficacy of antiparasitic treatments [120]. Careful prescriptions and diet formulations are observed in 'captivity' yet the prevalence of enteric helminths and protozoa was also higher in this context than in 'pet' monkeys, suggesting stronger effects of environmental exposure and opportunity for reinfection in the prevalence of these infections.

It is surprising that despite preventive measures applied at government-regulated captive facilities, neither parasite prevalence nor richness were lower in the 'captivity' context (Figs 2 and 4). The zoos and rescue centers in our study followed similar practices to prevent infections and reduce parasite loads: quarantine upon arrival; preventive medication during quarantine and annual medical checks; and isolation, medical care, and treatment of clinically ill animals. Pathogen screenings were not consistent among facilities and totally absent in many of them, due largely to a lack of resources to implement them [35]. Our results indicate that the preventive measures applied at the time of our study were insufficient at limiting common zoonotic infections and needed to be reinforced.

Microbiome and parasite communities of free-ranging monkey species can be almost entirely replaced by newly acquired infections in captive settings [21,22]. Because this is a progressive process, it is important to consider the life-history of monkeys. Trafficked monkeys are often captured at an early age and most pets are surrendered when they reach adult size or sexual maturity [34]. This was reflected in a larger proportion of adults in 'captivity' (64%) than among 'pet' (30%) and 'trade' monkeys (27%). However, despite the presumed longer time since the moment of capture of most monkeys at zoos and rescue centers, and their previous passage through at least one of the other two contexts, parasite communities were not entirely nested (Tables 3 and S2). The higher divergence of parasite communities found in 'captivity' versus those found in monkeys at the 'pet' and 'trade' contexts confirms that although some infections are lost due to environmental changes and medical treatments, others are acquired within captive facilities [22,121]. *Campylobacter sp.*, *Shigella sonnei*, *Trichuris sp.*, and *Tricomonas sp.* were exclusively found in 'captivity'. We provide a list of parasites circulating in captive monkeys in Peru that could help facilities to improve preventive measures during quarantine (Table 3). We also highlight the need for more strict hygiene and decontamination of enclosures to reduce the spread of enteric parasites and their transmission between human and monkey residents at both zoos and rescue centers.

## Contributing factors to parasite richness

Parasite richness (PR) in free-ranging monkeys is influenced by various factors, including host phylogeny, geographical distribution, life-history traits, and spatial and social dynamics within the common ecological niche that support both host and parasite communities [2–4,77,122]. We found that taxonomic identity and geographic location at the time of sampling where the main contributing factors to PR in captive monkeys that originated from primate trafficking in Peru. Using taxonomic genera as a conservative approach for taxonomic identity, we found that squirrel (*Saimiri sp.*) and wooly monkeys (*Lagothrix sp.*) had significantly larger PR than tufted capuchins (*Sapajus sp.)* and tamarins (*Saguinus sp.*, *Leontocebus sp.*). Squirrel monkeys and tamarins are small species (adult body weight <1.2Kg) with overlapping distributions

within the lowland Amazon, similar foraging habits, and often found in disturbed forests around human settlements [123]. They are also treated similarly in the trade, being captured as young or adult individuals, and transported and sold in large groups [34]. Nevertheless, we found that squirrel monkeys harbored at least 21 parasite taxa, whereas tamarins had only six (Table 2), resulting in an adjusted PR 2.3 higher in squirrel monkeys. These findings suggest that host taxonomy has a stronger effect in PR than the ecological conditions and trafficking practices associated with the host. The zoonotic parasites identified in this study show an apparent higher host affinity for squirrel monkeys and wooly monkeys. While further research with larger sample sizes is necessary to confirm this effect, previous studies conducted in Peru on captive and free-ranging monkeys have reported similar trends. Higher prevalence and intensity of filarial infections were observed in squirrel monkeys, wooly monkeys, and gracile capuchins when compared to other free-ranging monkeys hunted for subsistence in the northern Peruvian Amazon [45]. Squirrel monkeys, wooly monkeys, and gracile capuchins showed a higher richness of enteric helminths and protozoa among monkey species evaluated at a Peruvian zoo [124]. Similarly, infections with enteric helminths and protozoa were more frequent in free-ranging wooly monkeys than in owl monkeys sharing the same forest patch [57]. Apart from taxonomic genus, other host characteristics (i.e., sex and age) did not have an effect in our PR model.

Geographic or spatial co-occurrence is another important factor affecting parasite sharing, and thus we expected monkeys at the same location would have similar opportunities to acquire zoonotic infections by sharing common enclosures, husbandry, and environmental conditions. We found spatial heterogeneity in PR, which was better represented by the city where animals were sampled than by the sampling event (S4 Table) suggesting a stronger effect of environmental conditions over husbandry practices and shared enclosures. However, we found the highest PR in Yurimaguas, and the lowest in Iquitos, two Amazon cities with similar climatic and epidemiological characteristics. Further characterization of environmental variables affecting PR in our system is needed to identify the source of heterogeneity between cities within the same geographic region.

## Study limitations

Monkey genera that are rarely found in captivity (e.g. within the *Pitheciidae* family) or difficult to sample (e.g., *Callithrix sp.*, *Aotus sp.)* are underrepresented in our study population or not equally represented across contexts. Though this prevented robust prevalence estimations, our study includes 12 out of the 13 genera of monkeys reportedly trafficked in Peru and provides good geographic coverage at the national level. Our prevalence and richness estimations are conservative. Not all parasite identities were confirmed by molecular methods (Table 1) and more than one parasite species may be represented within genus-level identifications (e.g., *Dipetalonema sp.)*. In addition, the exclusion of lethargic and debilitated individuals may have biased our results by excluding symptomatic, diseased animals from our sample, especially if these were more likely to occur in a specific context. Exclusions were infrequent, but we did not keep a record of them and cannot estimate the magnitude of this bias. Furthermore, lacking comprehensive data on parasite diversity across free-ranging monkey species and geographical regions in Peru, we cannot ascertain if parasite richness is reduced or exacerbated in captivity. It is, however, unlikely that the breadth of zoonotic infections we describe are reflecting only natural infections from free-ranging primates. We expect Table 3 and the list of host-parasite associations detailed in our metadata would serve as a baseline for future in-depth studies of the diversity and directionality of infections supported by trafficking Neotropical monkeys.

## Recommendations

We call for urgent action against wildlife trafficking and ownership of monkeys as pets. In Latin America, monkeys are sold as pets within the wildlife subsection of traditional food markets [8,34]. Though safe food provision is a function of these markets that must be protected, live wildlife sales do not respond to an essential need and as we demonstrate in this study, represent a source for food and environmental contamination. Risk reduction at markets would benefit from the promotion and enforcement of better water, sanitation, and hygiene (WASH) standards [118], requiring the removal of live wildlife sales to reduce potential contamination of raw meats and other foodstuff.

But not all trafficking occurs in local markets. Monkeys are among the most trafficked mammals worldwide. In the United States alone, there are about 15,000 pet monkeys of which up to 65% may correspond to platyrrhine monkeys [125]. Yet despite their increasing popularity in legal and illegal trade channels, there are no nation-wide regulations pertaining to primate pet keeping or sales across the United States [126]. In the United Kingdom, squirrel monkeys, tamarins, and marmosets are commonly kept as pets, and there are currently no prohibitions on keeping any primates as pets [126,127]. The lack of enforcement regarding captive-bred origin often results in 'legal' pets imported from countries where legal extraction and trade are not feasible [128,129]. Implementing bans on pet monkey ownership would demonstrate the commitment of consumer countries to curbing wildlife trafficking while safeguarding One Health. We have identified several pathogens of public health concern in trafficked monkeys that could be introduced into other regions and represent a health hazard for households keeping pet monkeys in both source and destination countries.

## Supporting information

**S1 Fig. Prevalence of *Mycobacterium tuberculosis* complex (MTBC) and Simian Foamyvirus (SFV) in captive monkeys found at each context for animal-human interaction in Peru.** Bar plot showing the proportion of monkeys with positive status for MTBC and SFV at each context.
(TIF)

**S2 Fig. Prevalence of hemoparasites in captive monkeys found at each context for animal-human interaction in Peru.** Bar plot showing the proportion of monkeys with positive status for *Trypanosoma sp*., *Mansonella sp*., *tryopanosoma cruzi*, *Dipetalonema sp*., and *Plasmodium malaria/brasilianum* and SFV across contexts.
(TIF)

**S3 Fig. Prevalence of enteric bacteria in captive monkeys found at each context for animal-human interaction in Peru.** Bar plot showing the proportion of monkeys with positive status for *Aeromonas sp*., *Aeromonas caviae*, *Aeromonas sobria*, *Aeromonas hydrophila*, *Campylobacter sp*., *Campylobacter jejunii*, *Campylobacter coli*, *Plesiomonas shigelloides*., *Salmonella sp*., *Shigella boydii*, *Shigella flexneri*, and *Shigella sonnei* across contexts.
(TIF)

**S4 Fig. Prevalence of enteric helminths and protozoa in captive monkeys found at each context for animal-human interaction in Peru.** Bar plot showing the proportion of monkeys with positive status for *Prostenorchis sp*., *Strongyloides sp*., hookworms, *Trichuris sp*., *Ascaris sp*., *Molineus sp*., *Balantidium sp*., *Giardia sp*., *Blastocystis sp*., *Cryptosporidium sp*., and *Entamoeba sp*. across contexts.
(TIF)

**S5 Fig. Prevalence of trichomonads in captive monkeys found at each context for animal-human interaction in Peru.** Bar plot showing the proportion of monkeys with positive status for *Dientamoeba sp*. and *Trichomonas sp*. in the trade and at captivity contexts.
(TIF)

**S1 Table. Frequency (Freq.) and prevalence (Prev.) of parasites in captive monkeys found at each context for animal-human interaction in Peru.** This table shows the frequency of detection and prevalence of each parasite type (MTBC, SFV, hemoparasites, enteric bacteria, enteric helminths, enteric protozoa, and trichomonads) among the contexts for animal-human interaction (captivity, pet, and trade) in which trafficked monkeys are found in Peru and the results of the chi-squared test comparing the homogeneity of proportions between contexts.
(DOCX)

**S2 Table. Pairwise dissimilarities of parasite communities between contexts for animal-human interaction and host genera of trafficked monkeys.** This table shows the Sorensen dissimilarity indexes obtained through pairwise comparisons of the parasite communities composition found at different context and different monkey genera.
(DOCX)

**S3 Table. Factor loadings of the Principal Components (PC) Analysis.** This table shows the correlation of the different parasite genera with the main two principal components explaining the variation between parasite community composition across contexts for animal-human interaction and host genera of trafficked primates in Peru.
(DOCX)

**S4 Table. Model selection results for parasite richness among captive primates in Peru.** This table summarizes the generalized linear models (GLM) and generalized linear mixed effects models (GLMM) built to evaluate the contribution of population characteristics to parasite richness. Models ranked by Akaike's information criterion with small-sample correction (AICc). Statistics include number of parameters (K), log-likelihood ($-2LL$), difference between AICc of each model and the best model ($\Delta$AICc), and evidence ratio (wi/w1). Models listed under each heading are included in the 95% confidence set.
(DOCX)

## Acknowledgments

We thank all the people that participated in field sampling, data collection, and laboratory analysis. Our special recognition to wildlife authorities and the staff at zoos and rescue centres who tirelessly work to reduce the impacts of primate trafficking in Peru.

## Author Contributions

**Conceptualization:** A. Patricia Mendoza, Bruno M. Ghersi, Micaela De La Puente, Nancy Cavero, Yohani Ibañez, Alberto Perez, Marcela Uhart, Sarah H. Olson, Marieke H. Rosenbaum.

**Data curation:** A. Patricia Mendoza, Ana Muñoz-Maceda, Bruno M. Ghersi, Micaela De La Puente, Carlos Zariquiey, Nancy Cavero, Yohani Ibañez, Janine Robinson, Sarah H. Olson, Marieke H. Rosenbaum.

**Formal analysis:** A. Patricia Mendoza, Ana Muñoz-Maceda, Bruno M. Ghersi, Micaela De La Puente, Janine Robinson, Sarah H. Olson, Marieke H. Rosenbaum.

**Funding acquisition:** Patricia G. Parker, Marcela Uhart, Sarah H. Olson, Marieke H. Rosenbaum.

**Investigation:** A. Patricia Mendoza, Bruno M. Ghersi, Micaela De La Puente, Carlos Zariquiey, Nancy Cavero, Yovana Murillo, Miguel Sebastian, Yohani Ibañez, Alberto Perez, Marcela Uhart, Janine Robinson, Sarah H. Olson, Marieke H. Rosenbaum.

**Methodology:** A. Patricia Mendoza, Ana Muñoz-Maceda, Bruno M. Ghersi, Micaela De La Puente, Carlos Zariquiey, Nancy Cavero, Miguel Sebastian, Yohani Ibañez, Alberto Perez, Marcela Uhart, Janine Robinson, Sarah H. Olson, Marieke H. Rosenbaum.

**Project administration:** A. Patricia Mendoza, Nancy Cavero, Yovana Murillo, Patricia G. Parker, Alberto Perez, Marcela Uhart.

**Resources:** Miguel Sebastian, Yohani Ibañez, Patricia G. Parker, Alberto Perez, Marcela Uhart, Janine Robinson, Sarah H. Olson, Marieke H. Rosenbaum.

**Software:** Ana Muñoz-Maceda.

**Supervision:** A. Patricia Mendoza, Nancy Cavero, Yovana Murillo, Patricia G. Parker, Alberto Perez, Marcela Uhart, Janine Robinson, Sarah H. Olson.

**Validation:** A. Patricia Mendoza, Nancy Cavero, Alberto Perez, Marcela Uhart.

**Visualization:** A. Patricia Mendoza, Bruno M. Ghersi.

**Writing – original draft:** A. Patricia Mendoza, Ana Muñoz-Maceda, Sarah H. Olson, Marieke H. Rosenbaum.

**Writing – review & editing:** A. Patricia Mendoza, Ana Muñoz-Maceda, Bruno M. Ghersi, Micaela De La Puente, Carlos Zariquiey, Nancy Cavero, Yovana Murillo, Miguel Sebastian, Yohani Ibañez, Patricia G. Parker, Alberto Perez, Marcela Uhart, Janine Robinson, Sarah H. Olson, Marieke H. Rosenbaum.

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
