## [Decision Letter · Decision Letter 0]

19 Apr 2023

PONE-D-23-07657Diversity and prevalence of zoonotic infections at the animal-human interface generated by primate trafficking in PeruPLOS ONE

Dear Dr. Mendoza Becerra,

Thank you for submitting your manuscript to PLOS ONE. After careful consideration, we feel that it has merit but does not fully meet PLOS ONE’s publication criteria as it currently stands. Therefore, we invite you to submit a revised version of the manuscript that addresses the points raised during the review process. Please, pay attention to the comments and suggestions made by Reviewer #3. English should be revised.

We look forward to receiving your revised manuscript.

Kind regards,

M. Andreína Pacheco, Ph.D.

Academic Editor

PLOS ONE

Journal Requirements: 

3. We note that Figure (1) in your submission contain copyrighted images. All PLOS content is published under the Creative Commons Attribution License (CC BY 4.0), which means that the manuscript, images, and Supporting Information files will be freely available online, and any third party is permitted to access, download, copy, distribute, and use these materials in any way, even commercially, with proper attribution. For more information, see our copyright guidelines: http://journals.plos.org/plosone/s/licenses-and-copyright.

1. You may seek permission from the original copyright holder of Figure(s) [#] to publish the content specifically under the CC BY 4.0 license. 

Reviewers' comments:

Reviewer's Responses to Questions

**Comments to the Author**

1. Is the manuscript technically sound, and do the data support the conclusions?

Reviewer #1: Yes

Reviewer #2: Yes

Reviewer #3: Yes

2. Has the statistical analysis been performed appropriately and rigorously? 

Reviewer #1: Yes

Reviewer #2: Yes

Reviewer #3: Yes

3. Have the authors made all data underlying the findings in their manuscript fully available?

Reviewer #1: Yes

Reviewer #2: Yes

Reviewer #3: Yes

4. Is the manuscript presented in an intelligible fashion and written in standard English?

Reviewer #1: Yes

Reviewer #2: Yes

Reviewer #3: No

5. Review Comments to the Author

Reviewer #1: It is an original contribution, the document should define and clarify the wording according to the type of parasite and it is evident that in the wording there is a mixture of parasite types and taxonomic categories.

In Table 2, in the values of total samples per genus, primate species, the samples analysed per species should be presented, avoiding presenting the results as total samples per genus.

The discussion is encouraged to be based on the analysis of prevalences and diversities by species and not only by primate genus.

Reviewer #2: This is a remarkable study of the zoonotic parasites in trafficked primates in Peru. It is an outstanding sampling, diagnosis and analytical effort. The manuscript is clear, well written, organized, an the results are highly relevant for primate parasitology, primate conservation and public health; thus, I suggest publication. There are just a few minor details that I enlist below:

Line 71: maybe “infectious diseases”

Line 74: since authors are talking about parasite richness I recommend “number of parasite taxa” to avoid confusion with abundance estimates.

Lines 179 -180: “parasite taxa” referring to my previous comment.

Line 199: maybe mentioned the R packages employed.

Lines 218 and 222: “31 parasite taxa”, and “32 parasite taxa”. To avoid confusion with abundance

Lines 294-313: It is important to mention somewhere in this paragraph, that even though you found in monkeys parasites that can also infect humans, the direction of transmission either monkey-human or human-monkey remains to be determine. Because, as you mentioned later in the discussion, what prevents a monkey to get infected by a human parasite from their captor or owner or keeper?

Finally, I strongly suggest the authors to define either in the introduction or in methods the way their employing the term “zoonotic”. It is clear to me that they refer to transmission in both directions; but nowadays this term is used in different contexts, for example, the WHO define it as “an infectious disease that has jumped from a non-human animal to humans”, while the CDC states “Zoonotic diseases (also known as zoonoses) are caused by germs that spread between animals and people”. Thus a clear definition would guide the reader to the approach that authors are using in the study

Reviewer #3: This manuscript is aimed to compilate the information on pathogens infecting non-

human primates from three different contexts (captivity, households, and trade) in

nine locations in Peru, through a cross-sectional study carried out between 2010

and 2012.

The subject of the manuscript is relevant on the primate parasitology topic and in

the One-Health perspective. This manuscript fully corresponds to the scope of the

journal.

I consider that the manuscript needs some modifications, clarifications, and

corrections before being considered for publication, as well as a general English

revision. Some commentaries and suggestions are described below:

Title

The title states “primate trafficking”, however, according to the methods also

scenarios of captivity and pet were screened. Thus, I suggest modifying the title to

make it more accurate.

Abstract

- Line 35: “32 parasite taxa” instead of “32 parasites”.

- Line 38: as not all parasites were identified at the species level, I would suggest

changing “parasite species richness” to “parasite richness”/ “parasite taxa richness”

along the manuscript.

- Line 43: “humans” instead of “human”.

-Please include some details about the methods used for parasite identification

(e.g. light microscopy, bacteriological culture, and PCR), and type of samples

collected.

Key words

- According to the international primatological community, the term “Neotropical

primates” should be avoided because of the colonial overtones. Please modify it.

Introduction

- Line 78: delete “all of them platyrrhine monkeys”.

- Line 88: 320 parasite species? Parasite taxa? Please specify.

- Include abbreviation for “non-human primates” along the document.

- Please include some specific details about previous reports of parasites infecting

non-human primates in Peru. If available, also include the parasite’s prevalence, as

it is a topic dealt with in this manuscript.

Methods

- Lines 118-122: this information is more suitable for the introduction section.

- Table 1: “Dientamoeba” instead of “Dientomoeba”

- Line 145: “thin” instead of “think”

- Line 159: it is stated only “captive monkeys”, please specify if you are referring

also to the other animal-human contexts.

- Line 169: “for each parasite taxa” instead of “for each parasite”

- Line 174: “number of parasite taxa” instead of “number of parasites”.

- Line 184: “context” instead of “contexts”

Results

- Line 208: “including” instead of “covering”

- Table 2: “cassiquiarensis” instead of “cassiquairensis”, “parasite richness”/

“parasite taxa richness” instead of “parasite species richness (PSR)”.

- Line 222: “parasitesº”? or parasite taxa?. Please confirm.

- Table 3: Please modify the terms and abbreviations for “OWM: Old World

Monkeys” and “NP: Neotropical primates”

- Line 260: “nine parasite taxa” instead of “nine parasites”.

Discussion

- Line 288: please modify this sentence “widespread zoonotic infections are a daily

threat” as the sampling was conducted more than ten years ago, the actual

situation could be different.

- Lines 294-297: this information is more suitable for the introduction section.

- Line 403: “parasite taxa” instead of “parasite species”.

- Lines 393-403: please add some information on previously reported parasites

infecting free-ranging monkeys in Peru.

- Line 448: “identified” instead of “identify”.

6. PLOS authors have the option to publish the peer review history of their article (what does this mean?). If published, this will include your full peer review and any attached files.

Reviewer #1: No

Reviewer #2: No

Reviewer #3: No

---

## [Author Response · Author response to Decision Letter 0]

7 Jun 2023

Reviewer #1:

It is an original contribution, the document should define and clarify the wording according to the type of parasite and it is evident that in the wording there is a mixture of parasite types and taxonomic categories.

We thank the reviewer for the careful reading of our manuscript. We have clarified the wording and standardized the terms along the manuscript.

The use of the term “parasite” to represent different parasite types is defined at the beginning of the Statistical analysis section of the Methodology (lines 167-171) and reads as follows: 

“All the infectious agents we tested for are referred to as ‘parasites’ because the presence of clinical disease was not investigated (63). Parasite prevalence was estimated as the number of infected individuals over the total number of individuals evaluated for each parasite taxa and parasite type (i.e., mycobacteria, viruses, hemoparasites, enteric bacteria, enteric helminths, enteric protozoa, and trichomonads).”

As in the second sentence of the above paragraph, when pertaining, the use of “parasite” has been replaced by “parasite taxa”

In Table 2, in the values of total samples per genus, primate species, the samples analysed per species should be presented, avoiding presenting the results as total samples per genus.

The number of samples analyzed per species has been added into Table 2.

The discussion is encouraged to be based on the analysis of prevalences and diversities by species and not only by primate genus.

We appreciate this suggestion. However, the discussion of prevalence and diversity at species level falls outside the goals of the manuscript. Please note that the taxonomy of South American primates is changing, and some former species now correspond to multiple taxa. For example, species previously identified as Cebus albifrons in Peru now correspond to whether Cebus yuracus, Cebus cuscinus, or Cebus unicolor. We analyze the results by primate genus as it provides the highest level of taxonomic certainty in our study population. 

Reviewer #2:

This is a remarkable study of the zoonotic parasites in trafficked primates in Peru. It is an outstanding sampling, diagnosis and analytical effort. The manuscript is clear, well written, organized, an the results are highly relevant for primate parasitology, primate conservation and public health; thus, I suggest publication. There are just a few minor details that I enlist below:

We thank the reviewer for the careful reading of our manuscript, for considering it relevant, and for their time in providing detailed suggestions. We respond to each of their suggestions below:

Line 71: maybe “infectious diseases”

The term has been changed. The text now reads: 

“Non-human primates are also susceptible to infectious disease caused by human-associated parasites.”

Line 74: since authors are talking about parasite richness I recommend “number of parasite taxa” to avoid confusion with abundance estimates.

Parasite species richness (PSR) has been replaced by “parasite richness (PR)” along the manuscript.

Lines 179 -180: “parasite taxa” referring to my previous comment.

The term has been included. The text now reads: 

“The number of parasite taxa detected by each test was used to estimate the ‘maximum possible detection’ (MPD). MPD = the number of parasite taxa that could have been detected by all the tests carried out on samples from the same monkey.”

Line 199: maybe mentioned the R packages employed.

The main R packages employed have been cited. The text now reads:

“All statistical analyses were performed using R 4.0.3 (R Development Core Team, 2021), RStudio (RStudio Team, 2023) the vegan (66), the DescTools (67), the betapart (65), the factoextra (68), the glmulti (69), and the lmer4 (70) packages.”

Lines 218 and 222: “31 parasite taxa”, and “32 parasite taxa”. To avoid confusion with abundance

The term has been replaced. The text now reads: 

“Up to seven tests were carried out on samples from the same monkey (median number of tests=3.5, IQR=2.0-5.0) reaching a maximum possible detection of 31 parasite taxa (median MPD=17, IQR=13.0-24.0). […] A total of 1,313 tests were performed, resulting in the detection of 32 parasite taxa.”

Lines 294-313: It is important to mention somewhere in this paragraph, that even though you found in monkeys parasites that can also infect humans, the direction of transmission either monkey-human or human-monkey remains to be determine. Because, as you mentioned later in the discussion, what prevents a monkey to get infected by a human parasite from their captor or owner or keeper?

We thank the reviewer for this important observation. Bidirectional transmission of zoonotic parasites is now mentioned in the paragraph and reads as follows: 

“At markets, live monkeys are often observed in the proximity of poultry and raw meat, while pet monkeys are often in close direct contact with children and vulnerable adults within the family environment (8,9). Although infected monkeys may have acquired zoonotic parasites from humans or other infected animals while in captivity, their presence in markets and households represents an additional source of contamination and exposure to zoonotic pathogens at these scenarios.”

Finally, I strongly suggest the authors to define either in the introduction or in methods the way their employing the term “zoonotic”. It is clear to me that they refer to transmission in both directions; but nowadays this term is used in different contexts, for example, the WHO define it as “an infectious disease that has jumped from a non-human animal to humans”, while the CDC states “Zoonotic diseases (also known as zoonoses) are caused by germs that spread between animals and people”. Thus a clear definition would guide the reader to the approach that authors are using in the study

We appreciate this important suggestion. The definition of zoonoses has been included in the abstract and first paragraph of the introduction, and reads as follows:

“Of highest concern among infections that can spread through wildlife trade and traffic are the zoonoses – those infections caused by parasites that can be transmitted between humans and other animals.”

 

Reviewer #3:

This manuscript is aimed to compilate the information on pathogens infecting non-human primates from three different contexts (captivity, households, and trade) in nine locations in Peru, through a cross-sectional study carried out between 2010 and 2012. The subject of the manuscript is relevant on the primate parasitology topic and in the One-Health perspective. This manuscript fully corresponds to the scope of the journal. I consider that the manuscript needs some modifications, clarifications, and corrections before being considered for publication, as well as a general English revision. Some commentaries and suggestions are described below:

We thank the reviewer for its careful reading of the manuscript and their suggestions to improve it. We respond to their observations below:

Title

The title states “primate trafficking”, however, according to the methods also

scenarios of captivity and pet were screened. Thus, I suggest modifying the title to

make it more accurate.

We thank the reviewer for this important suggestion. As indicated in the methodology, our study included only wild-caught primates, which originate in the traffic. 

The origin of pet and captive monkeys in the trade is also described in the third paragraph of the introduction, which reads: 

“Monkeys offered for sale or obtained illegally as pets are frequently confiscated by local authorities, temporarily held in custody facilities, and ultimately euthanized or transferred to government-regulated zoos and rescue centers”

Traffic origin is also mentioned in the first paragraph of the Methods, under Study design and sampling strategy, which reads:

“We sampled wild-caught monkeys at zoos, rescue centers, households, wildlife markets, and temporary custody facilities and classified them into three distinct ‘contexts’ where animal-human interaction occurs: 1) captivity (zoos and rescue centers); 2) pet (households); and 3) trade (wildlife markets or seized during transport to commercial establishments). Monkeys confiscated by local authorities or voluntarily surrendered were assigned to the context they came from (trade or pet) and sampled within the first week of their arrival at the custody facility. To ensure that only animals originated in the primate trafficking were represented in the study, monkeys born in captivity were excluded.”

Abstract

- Line 35: “32 parasite taxa” instead of “32 parasites”.

The term has been replaced, now the text reads:

“We detected 32 parasite taxa including…”

- Line 38: as not all parasites were identified at the species level, I would suggest

changing “parasite species richness” to “parasite richness”/ “parasite taxa richness”

along the manuscript.

Parasite species richness (PSR) has been replaced by “parasite richness (PR)” along the manuscript.

- Line 43: “humans” instead of “human”.

The correction has been made. The text now reads:

“Our findings illustrate that the threats of wildlife trafficking to One Health encompass exposure to multiple zoonotic parasites well-known to cause disease in humans, monkeys, and other species.”

-Please include some details about the methods used for parasite identification

(e.g. light microscopy, bacteriological culture, and PCR), and type of samples

collected.

Laboratory techniques are listed in Table 1. Detailed procedures for each technique can be found in the publications referenced also in Table 1. 

Key words

- According to the international primatological community, the term “Neotropical

primates” should be avoided because of the colonial overtones. Please modify it.

Introduction

Although we disagree with the reviewer, the term Neotropical has been removed from most of the manuscript. Please note we used Neotropical as reference to the geographic region, the Neotropics. We avoided the use of “American primates” as it may be misleading for some readers and have used the term “platyrrhine monkeys” although we recognize readers without a primatological background may be unfamiliar with primate parvorders. 

- Line 78: delete “all of them platyrrhine monkeys”.

The phrase has been deleted.

- Line 88: 320 parasite species? Parasite taxa? Please specify.

The term “parasites” has been replaced by “parasite taxa”

- Include abbreviation for “non-human primates” along the document.

The abbreviation has been included in Line 69. Now it reads: 

“As a result of their interaction with us, captive non-human primates (NHP) acquire human-associated parasites…”

- Please include some specific details about previous reports of parasites infecting

non-human primates in Peru. If available, also include the parasite’s prevalence, as

it is a topic dealt with in this manuscript.

We have included information about parasites previously reported in Peru and some prevalence data in the introduction. The fourth paragraph of the introduction now reads: 

“More than 320 parasite taxa have been reported in monkeys across the Americas, encompassing various types such as enteric helminths, enteric protozoa, hemoparasites, viruses, bacteria, and ectoparasites (40–44). Among these, at least 74 taxa are known to infect humans (42,43). Focusing on zoonotic or potentially zoonotic parasites, studies in Peruvian wild and captive monkeys have detected infections with hemoparasites belonging to the genera Trypanosoma, Plasmodium, Dipetalonema, and Mansonella (45–51), enteric helminths such as Ancyclostoma sp., Ascaris sp., Strongyloides spp., Trypanoxiuris sp., Trichuris trichiura, and Schistosoma mansoni. (52–55), and enteric protozoa of the genera Blastocytis, Balantidium, Cryptosporidium, Coccidia, Eimeria and Entamoeba (54–57). In the northern Peruvian Amazon, the zoonotic bacteria Campylobacter spp. were reported in 21% wild monkeys and 32% pet monkeys (58), while primate rescue centers in the same region report primates infected with antimicrobial-resistant enterobacteria such as Escherichia coli, Salmonella arizonae, Shigella sp., and Serratia spp. (41). Within the country, human-associated infections such as mycobacteria (59) and human herpesvirus type 1 (60) have been documented exclusively in captive primates, whereas a high seroprevalence of arbovirus antibodies has been observed solely in wild monkeys (61,62).”

Methods

- Lines 118-122: this information is more suitable for the introduction section.

We appreciate the suggestion. The following text has been moved from methods to the third paragraph of the introduction: 

“Monkeys offered for sale or obtained illegally as pets are frequently confiscated by local authorities, temporarily held in custody facilities, and ultimately euthanized or transferred to government-regulated zoos and rescue centers (34,35)”

- Table 1: “Dientamoeba” instead of “Dientomoeba”

The term has been corrected.

- Line 145: “thin” instead of “think”

The term has been corrected.

- Line 159: it is stated only “captive monkeys”, please specify if you are referring

also to the other animal-human contexts.

The text has been modified and now reads: 

“Table 1. Sampling, storage, and laboratory technique used for the detection of zoonotic parasites in captive, pet, and traded monkeys in Peru”

- Line 169: “for each parasite taxa” instead of “for each parasite”

The term has been replaced, now the text reads:

“Parasite prevalence was estimated as the number of infected individuals over the total number of individuals evaluated for each parasite taxa and parasite type”

- Line 174: “number of parasite taxa” instead of “number of parasites”.

The term has been replaced, now the text reads:

“Parasite richness (PR) was estimated as the number of parasite taxa detected in each monkey by context…”

- Line 184: “context” instead of “contexts”

The term has been corrected.

Results

- Line 208: “including” instead of “covering”

The term has been replaced. Now it reads:

“Samples were obtained from 388 monkeys including 18 species and 12 genera…”

- Table 2: “cassiquiarensis” instead of “cassiquairensis”, “parasite richness”/

“parasite taxa richness” instead of “parasite species richness (PSR)”.

The terms have been corrected.

- Line 222: “parasitesº”? or parasite taxa?. Please confirm.

The term has been replaced. The text now reads: 

“Up to seven tests were carried out on samples from the same monkey (median number of tests=3.5, IQR=2.0-5.0) reaching a maximum possible detection of 31 parasite taxa (median MPD=17, IQR=13.0-24.0). […] A total of 1,313 tests were performed, resulting in the detection of 32 parasite taxa.”

- Table 3: Please modify the terms and abbreviations for “OWM: Old World

Monkeys” and “NP: Neotropical primates”

“OWM: Old World Monkeys” has been replaced by “catarrhines” and “NP: Neotropical primates” has been replaced by “platyrrhines” along the document.

- Line 260: “nine parasite taxa” instead of “nine parasites”.

The term has been replaced. The text now reads:

“A maximum of nine parasite taxa were detected in the same individual...”

Discussion

- Line 288: please modify this sentence “widespread zoonotic infections are a daily threat” as the sampling was conducted more than ten years ago, the actualsituation could be different.

The text now reads:

“Our results illustrate that well-known, widespread zoonotic infections can be a daily threat at the animal-human...”

- Lines 294-297: this information is more suitable for the introduction section.

The text has been removed from the discussion. 

- Line 403: “parasite taxa” instead of “parasite species”.

The term has been replaced. The text now reads: 

“at least 21 parasite taxa were found in squirrel monkeys…”

- Lines 393-403: please add some information on previously reported parasites

infecting free-ranging monkeys in Peru.

We thank the reviewer for such an important suggestion. References of previous studies in Peru have been added. The text now reads: 

“The zoonotic parasites identified in this study show an apparent higher host affinity for squirrel monkeys and wooly monkeys. While further research with larger sample sizes is necessary to confirm this effect, previous studies conducted in Peru on captive and free-ranging monkeys have reported similar trends. Higher prevalence and intensity of filarial infections were observed in squirrel monkeys, wooly monkeys, and gracile capuchins when compared to other free-ranging monkeys hunted for subsistence in the northern Peruvian Amazon (45). Squirrel monkeys, wooly monkeys, and gracile capuchins showed a higher richness of enteric helminths and protozoa among monkey species evaluated at a Peruvian zoo (124). Similarly, infections with enteric helminths and protozoa were more frequent in free-ranging wooly monkeys than in owl monkeys sharing the same forest patch (57). Apart from taxonomic genus, other host characteristics (i.e., sex and age) did not have an effect in our PR model.”

- Line 448: “identified” instead of “identify”.

The term has been replaced. The text now reads: 

“We identified several pathogens of public health relevance in trafficked monkeys…”

---

## [Decision Letter · Decision Letter 1]

17 Jul 2023

PONE-D-23-07657R1Diversity and prevalence of zoonotic infections at the animal-human interface of primate trafficking in PeruPLOS ONE

Dear Dr. Mendoza,

Thank you for submitting your manuscript to PLOS ONE. After careful consideration, we feel that it has merit but does not fully meet PLOS ONE’s publication criteria as it currently stands. Therefore, we invite you to submit a revised version of the manuscript that addresses the points raised during the review process.

We look forward to receiving your revised manuscript.

Kind regards,

Érika Martins Braga, Ph.D.

Academic Editor

PLOS ONE

Journal Requirements:

Additional Editor Comments:

The authors have satisfactorily addressed most of all review comments and made the necessary changes to the manuscript.

A particular aspect that warrants consideration is the scarcity of information regarding the prevalence of Plasmodium malariae/Plasmodium brasilianum infections on a global scale and in the Amazon basin, despite the anticipation of a low prevalence and minimal health impact. Therefore, it is essential to thoroughly explore this subject (see line 336), taking into account the zoonotic nature of these species but not other human malaria species such as Plasmodium falciparum and Plasmodium vixax.

Please, also correct Trypanosoma minasense in line 374.

Reviewers' comments:

Reviewer's Responses to Questions

**Comments to the Author**

1. If the authors have adequately addressed your comments raised in a previous round of review and you feel that this manuscript is now acceptable for publication, you may indicate that here to bypass the “Comments to the Author” section, enter your conflict of interest statement in the “Confidential to Editor” section, and submit your "Accept" recommendation.

Reviewer #2: All comments have been addressed

Reviewer #3: All comments have been addressed

2. Is the manuscript technically sound, and do the data support the conclusions?

Reviewer #2: Yes

Reviewer #3: (No Response)

3. Has the statistical analysis been performed appropriately and rigorously? 

Reviewer #2: Yes

Reviewer #3: (No Response)

4. Have the authors made all data underlying the findings in their manuscript fully available?

Reviewer #2: Yes

Reviewer #3: (No Response)

5. Is the manuscript presented in an intelligible fashion and written in standard English?

Reviewer #2: Yes

Reviewer #3: (No Response)

6. Review Comments to the Author

Reviewer #2: (No Response)

Reviewer #3: (No Response)

7. PLOS authors have the option to publish the peer review history of their article (what does this mean?). If published, this will include your full peer review and any attached files.

Reviewer #2: No

Reviewer #3: No

---

## [Author Response · Author response to Decision Letter 1]

31 Aug 2023

We thank the reviewer for the careful reading of our manuscript. We have included the information required and corrected the typo mentioned by the reviewer. 

Regarding the reviewer’s concern about further investigation in Plasmodium malariae/Plasmodium brasilianum – This is a topic that certainly requires further investigation. We are preparing a manuscript discussing the circulation of this parasite in captive facilities in Peru. The potential of this parasite for zoonotic transmission is discussed in more detail in that additional manuscript. The current manuscript was intended to go over the broader range of pathogens infecting captive primates, and so it has not expanded on malaria.

---

## [Editor Report · Decision Letter 2]

10 Sep 2023

Diversity and prevalence of zoonotic infections at the animal-human interface of primate trafficking in Peru

PONE-D-23-07657R2

Dear Dr. Mendoza,

We’re pleased to inform you that your manuscript has been judged scientifically suitable for publication and will be formally accepted for publication once it meets all outstanding technical requirements.

Kind regards,

Érika Martins Braga, Ph.D.

Academic Editor

PLOS ONE

---

## [Editor Report · Acceptance letter]

26 Jun 2023

PONE-D-23-07657R1 

Diversity and prevalence of zoonotic infections at the animal-human interface of primate trafficking in Peru 

Dear Dr. Mendoza:

I'm pleased to inform you that your manuscript has been deemed suitable for publication in PLOS ONE. Congratulations! Your manuscript is now with our production department. 

Kind regards, 

on behalf of

Dr. M. Andreína Pacheco 

Academic Editor

PLOS ONE